# SARS-CoV-2 positivity rates associated with circulating 25-hydroxyvitamin D levels

**Harvey W. Kaufman[1], Justin K. Niles[1], Martin H. Kroll[1], Caixia Bi[1], Michael F. Holick ID [2]***

**1** Medical Informatics, Quest Diagnostics, Secaucus, New Jersey, United States of America, **2** Department of Medicine, Boston University School of Medicine, Boston, Massachusetts, United States of America

* mfholick@bu.edu

**Data Availability Statement:** Data underlying the study cannot be made publicly available due to ethical concerns about patient confidentiality. Data will be made available to qualified researchers on request to HealthTrends@QuestDiagnostics.com.

## Abstract

Until treatment and vaccine for coronavirus disease-2019 (COVID-19) becomes widely available, other methods of reducing infection rates should be explored. This study used a retrospective, observational analysis of deidentified tests performed at a national clinical laboratory to determine if circulating 25-hydroxyvitamin D (25(OH)D) levels are associated with severe acute respiratory disease coronavirus 2 (SARS-CoV-2) positivity rates. Over 190,000 patients from all 50 states with SARS-CoV-2 results performed mid-March through mid-June, 2020 and matching 25(OH)D results from the preceding 12 months were included. Residential zip code data was required to match with US Census data and perform analyses of race/ethnicity proportions and latitude. A total of 191,779 patients were included (median age, 54 years [interquartile range 40.4–64.7]; 68% female. The SARS-CoV-2 positivity rate was 9.3% (95% C.I. 9.2–9.5%) and the mean seasonally adjusted 25(OH)D was 31.7 (SD 11.7). The SARS-CoV-2 positivity rate was higher in the 39,190 patients with "deficient" 25(OH)D values (<20 ng/mL) (12.5%, 95% C.I. 12.2–12.8%) than in the 27,870 patients with "adequate" values (30–34 ng/mL) (8.1%, 95% C.I. 7.8–8.4%) and the 12,321 patients with values ≥55 ng/mL (5.9%, 95% C.I. 5.5–6.4%). The association between 25(OH)D levels and SARS-CoV-2 positivity was best fitted by the weighted second-order polynomial regression, which indicated strong correlation in the total population ($R^2 = 0.96$) and in analyses stratified by all studied demographic factors. The association between lower SARS-CoV-2 positivity rates and higher circulating 25(OH)D levels remained significant in a multivariable logistic model adjusting for all included demographic factors (adjusted odds ratio 0.984 per ng/mL increment, 95% C.I. 0.983–0.986; p<0.001). SARS-CoV-2 positivity is strongly and inversely associated with circulating 25(OH)D levels, a relationship that persists across latitudes, races/ethnicities, both sexes, and age ranges. Our findings provide impetus to explore the role of vitamin D supplementation in reducing the risk for SARS-CoV-2 infection and COVID-19 disease.

**Funding:** Quest Diagnostics provided support in the form of salaries for authors JKN, BC, MHK, and HWK and consulting fees for MFH but did not have any additional role in the study design, data collection and analysis, decision to publish, or preparation of the manuscript. The specific roles of these authors are articulated in the 'author contributions' section.

**Competing interests:** HWK, JKN, MHK, and CB are employees of Quest Diagnostics. HWK, MHK and CB own stock in Quest Diagnostics. MFH is a consultant to Quest Diagnostics and was on the speakers' bureau for Abbott Inc. and Hyatt Pharmaceutical Industries Company PLC.

## Introduction

Studies suggest an association between vitamin D deficiency and risk of viral upper respiratory tract infections and mortality from coronavirus disease-2019 (COVID-19) [1, 2]. This relationship is anticipated, given that vitamin D has numerous actions affecting the innate and adaptive immune systems. Respiratory monocytes/macrophages and epithelial cells constitutively express the vitamin D receptor. Acting through this receptor, vitamin D may be important in protection against respiratory infections [3]. In addition, an important action of vitamin D is suppressing excessive cytokine release that can present as a "cytokine storm," a common cause of COVID-19-related morbidity and mortality [4]. The role of vitamin D supplementation in reducing the risk of infection by severe acute respiratory disease coronavirus-2 (SARS-CoV-2) has not been studied. Better understanding of the relation between vitamin D status and SARS-CoV-2 NAAT positivity rates is appropriate before evaluating this potential intervention.

Previous studies examined latitude-related differences in COVID-19 outcomes related to vitamin D [1, 5]. However, to our knowledge, only two studies investigated the direct relationship between vitamin D status and SARS-CoV-2 positivity, and these came to opposite conclusions [6, 7]. Both were based on small numbers of paired SARS-CoV-2 and 25(OH)D results, and neither involved US patients. In this study, we evaluated the association of circulating 25-hydroxyvitamin D [25(OH)D] levels, a measure of vitamin D status, with positivity for SARS-CoV-2 as assessed with nucleic acid amplification testing (NAAT).

## Methods

### Study population

In this retrospective, observational analysis of deidentified test results from a clinical laboratory, a Quest Diagnostics-wide unique patient identifier was used to match all results of SARS-CoV-2 testing performed March 9 through June 19, 2020, with 25(OH)D results from the preceding 12 months. Analysis was limited to one SARS-CoV-2 result per patient; patients were considered to have a positive SARS-CoV-2 result if any test result indicated positivity. When multiple 25(OH)D results were available, the most recent was selected. We excluded specimens with inconclusive results (one out of two SARS-CoV-2 targets detected) or missing residential zip code data, which are needed to assign race/ethnicity proportions and latitude.

### Laboratory methods

All SARS-CoV-2 RNA NAATs were performed by Quest Diagnostics using one of four United States Food and Drug Administration (FDA) Emergency Use Authorized tests (Quest Diagnostics SARS-CoV-2 RNA [COVID-19], Qualitative NAAT; Hologic Panther Fusion SARS--CoV-2 assay; Roche Diagnostics cobas® SARS-CoV-2 test; or Hologic Aptima SARS-CoV-2 assay). We combined results from all four tests due to their very similar sensitivity and specificity [8–11]. Total 25(OH)D was measured using a chemiluminescent immunoassay (DiaSorin LIAISON® XL 25-hydroxyvitamin D, total) or a laboratory-developed test based on liquid chromatograph/tandem mass spectrometry. The laboratory categorizes 25(OH)D results <20 ng/mL as deficient, 20–29 ng/mL as suboptimal, and ≥30 ng/mL as optimal. The laboratory assays are standardized and performed identically throughout Quest Diagnostics.

### Estimates by zip code

To analyze race/ethnicity, patient data were linked to estimated race/ethnicity proportions reported by zip code in the 2018 5-year American Community Survey (ACS) [12]. Zip codes

with estimated proportions of black non-Hispanic population over 50% are referred to as "predominately black non-Hispanic." The same pattern was followed for "predominantly Hispanic" and "predominantly white non-Hispanic" zip codes. Latitude for each zip code, acquired from SAS reference data, was stratified into three groups: >40 degrees ("northern"); 32–40 degrees ("central"); or <32 degrees ("southern").

### Vitamin D seasonality adjustment

We adjusted for vitamin D seasonality with a model based on a previous $25(OH)D_3$ study, utilizing Quest Diagnostics results that fit the present study data well [13].

### Statistical analyses

Comparisons of proportions were analyzed using the chi-square test. Comparisons of means were analyzed using the t-test. Concentrations of circulating 25(OH)D are reported in ng/mL. Values <20 ng/mL or ≥60 ng/mL were assigned a value of 19 ng/mL or 60 ng/mL, respectively. Age was stratified into two groups: <60 years and ≥60 years for convenience. The correlation between 25(OH)D values and SARS-CoV-2 positivity were fitted the best by the weighted second-order polynomial regression. For regressions of predominately black non-Hispanic and Hispanic zip codes, 25(OH)D values were grouped into bins with two values from 20–29 (20–21, 22–23, etc.), and bins with 5 values thereafter (30–34, 35–39, etc.), because of the relatively low number of patients with 25(OH)D values ≥30. For all other polynomial regressions, 25(OH)D values were grouped into bins with two values. Multivariable logistic regression was performed using a stepwise entry criterion of $p<0.05$, after excluding patients with missing values for any included factor. Analyses were performed using SAS Studio 3.6 on SAS 9.4 (SAS Institute) and R, version 3.6.1 (R Project for Statistical Computing). HIPAA clearly defines research use of data as analyzed for this and numerous other studies based on the Quest Diagnostics Data Informatics Warehouse (45 CFR 164.501, 164.508, 164.512(i) (See also 45 CFR 164.514(e), 164.528, 164.532) Quest Diagnostics takes the additional step of having its process reviewed annual by the Western Institutional Review Board (Puyallup, Washington) who has determined the process is "deemed exempt."

## Results

Our potential cohort included 218,372 patients. After excluding patients with missing residential zip code data (n = 26,387) or inconclusive SARS-CoV-2 NAAT results (n = 206), results from 191,779 (87.8%) patients remained for analysis. This cohort comprised patients from all 50 states and the District of Columbia. The median age and sex distribution of included and excluded patients were nearly identical: age, 54.0 years, IQR 40.4–64.7, vs. 53.7 years, IQR 39.7–64.5, and female, 68% vs. 67%, respectively. SARS-CoV-2 positivity was lower among included (9.3%, 95% C.I. 9.2–9.5%) than excluded (10.1%, 95% C.I. 9.7–10.4%) patients (p<0.001). 98.8% of included patients had 25(OH)D levels assessed with immunoassay testing methodology.

There was an association between lower SARS-CoV-2 positivity rates and higher circulating 25(OH)D levels (unadjusted odds ratio 0.979 per 1 ng/mL increment, 95% C.I. 0.977–0.980). Regression analysis indicated strong correlation (R-squared = 0.96) between 25(OH)D levels and SARS-CoV-2 positivity in the total population (Fig 1) and in northern, central, and southern latitudes (Fig 2A). The decrease in positivity rate associated with 25(OH)D levels appeared to plateau as values approached 55 ng/mL; SARS-CoV-2 positivity rates were similar between the 4,016 patients with values 55–59 ng/mL (6.0%, 95% C.I. 5.2–6.7%) and the 8,305 patients with higher values (5.9%, 95% C.I. 5.4–6.4%). The SARS-CoV-2 positivity rate was lower in

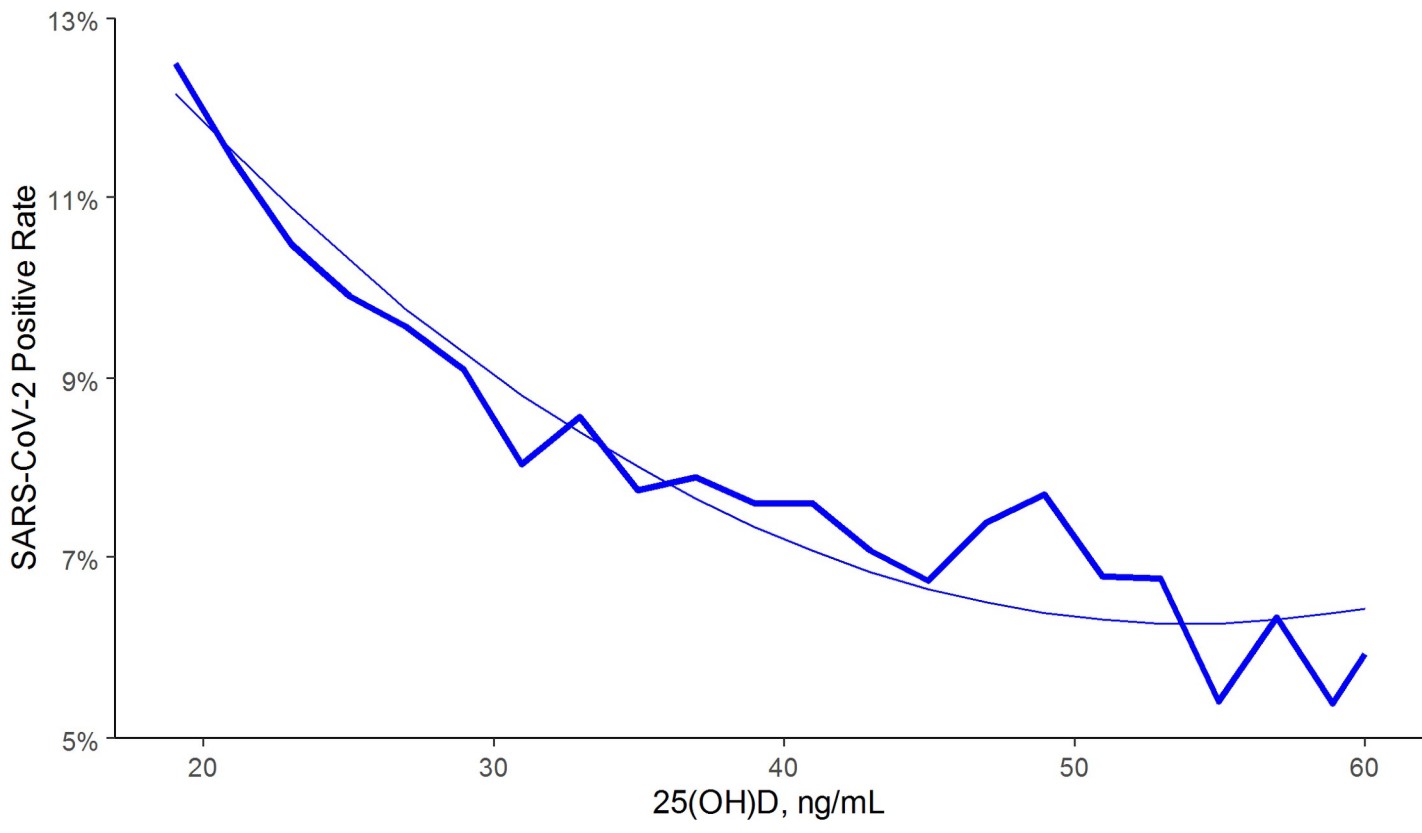

**Fig 1. SARS-CoV-2 NAAT positivity rates and circulating 25(OH)D levels in the total population.** Smooth line represents the weighted second order polynomial regression fit to the data associating circulating 25(OH)D levels (x) and SARS-CoV-2 positivity rates (y) where: $y = 0.2029 - 0.0052^*x + 4.8e\text{-}05^*x^2$; $R^2 = 0.96$. SI conversion factor: 1 ng/mL = 0.400641 nmol/L.

the 27,870 patients with "adequate" 25(OH)D values (30–34 ng/mL) (8.1%, 95% C.I. 7.8–8.4%) than in the 39,190 patients with "deficiency" (<20 ng/mL) (12.5%, 95% C.I. 12.2–12.8%) (difference 35%; p<0.001). Similarly, the SARS-CoV-2 positivity rate was lower in the 12,321 patients with 25(OH)D values ≥55 ng/mL (5.9%, 95% C.I. 5.5–6.4%) than in patients with adequate values (difference 27%; p<0.001).

SARS-CoV-2 positivity rates were higher in the 9,529 patients from predominately black non-Hispanic zip codes (15.7%, 95% C.I. 15.0–16.4%) and the 26,242 patients from predominately Hispanic zip codes (12.8%, 95% C.I. 12.4–13.2%) than in the 112,281 patients from predominately white non-Hispanic zip codes (7.2%, 95% C.I. 7.1–7.4%; p<0.001 for both comparisons). Mean (±SD) 25(OH)D levels were also higher in patients from predominately white non-Hispanic zip codes (33.0±11.9 ng/mL; 1 ng/mL = 0.400641 nmol/L) than in patients from predominately black non-Hispanic (29.1±11.0 ng/mL; p<0.001) or Hispanic (28.8±10.7 ng/mL; p<0.001) zip codes. Regression analysis indicated strong correlation between 25(OH) D levels and SARS-CoV-2 positivity in each of these groups (Fig 2B).

Compared to the 67,667 patients age ≥60 years, the 120,362 younger patients had significantly higher SARS-CoV-2 positivity (10.2%, 95% C.I. 10.0–10.3%, vs. 7.7%, 95% C.I. 7.5–7.9%; p<0.001) and lower mean 25(OH)D levels (29.4±10.8 ng/mL vs. 35.4±12.1 ng/mL; p<0.001). Compared to the 130,473 female patients, the 61,305 male patients had higher SARS-CoV-2 positivity (10.7% 95% C.I. 10.5–11.0% vs. 8.7%, 95% C.I. 8.5%-8.8%; p<0.001) and lower mean 25(OH)D levels (31.3±11.4 ng/mL vs. 31.9±11.8 ng/mL; p<0.001). Regression

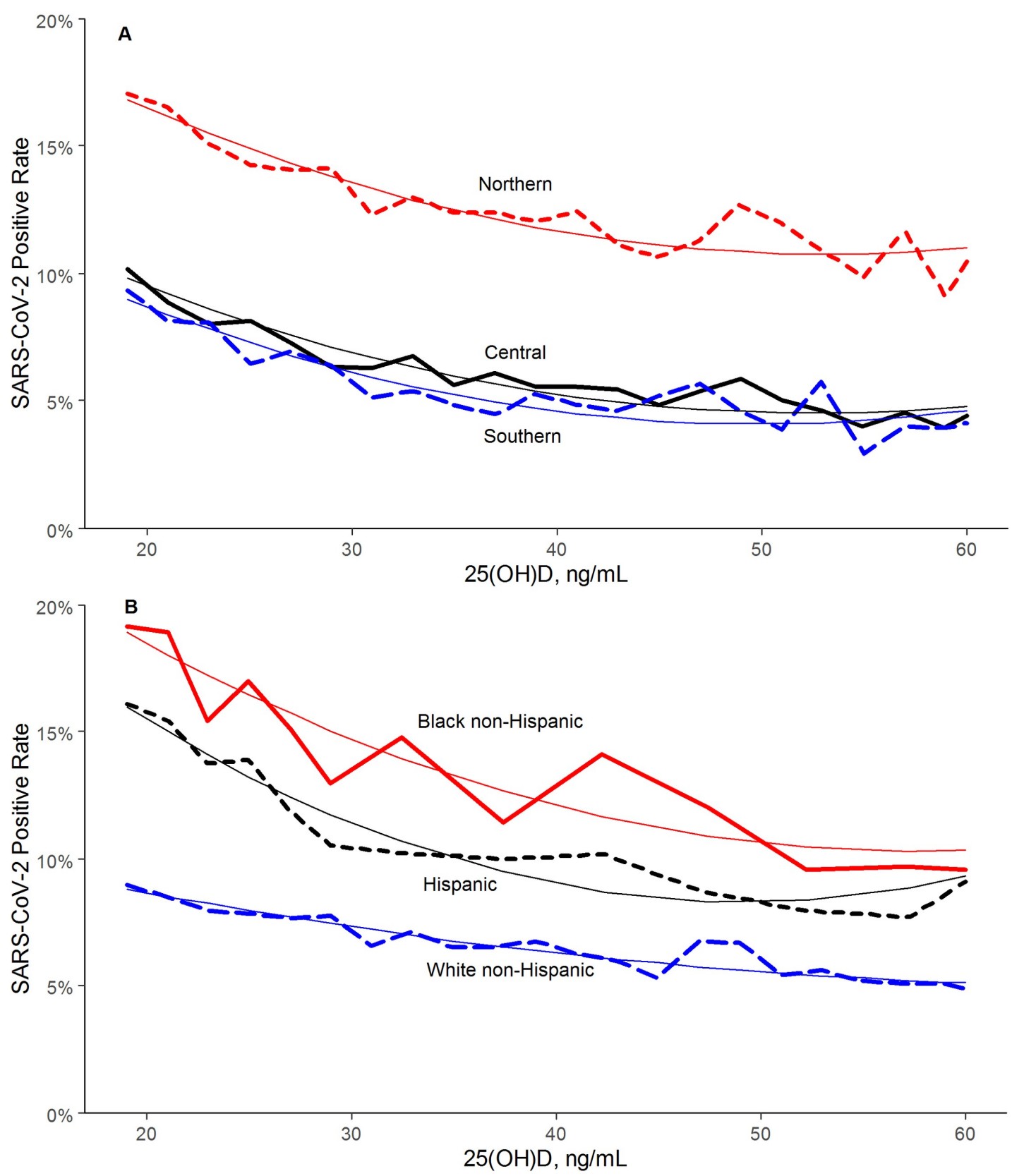

**Fig 2.** SARS-CoV-2 NAAT Positivity Rates and Circulating 25(OH)D Levels, (A) by Latitude Region and (B) Predominately Black non-Hispanic, Hispanic, and White non-Hispanic Zip Codes. Smooth lines represent the weighted second order polynomial regression fit to the data associating circulating 25(OH)D levels (x) and SARS-CoV-2 positivity rates (y) where: Northern: $y = 0.2544 - 0.0055^{*}x + 5.2e\text{-}05^{*}x^2$; $R^2 = 0.94$. Central: $y = 0.1745 - 0.0049^{*}x + 4.7e\text{-}05^{*}x^2$; $R^2 = 0.94$. Southern: $y = 0.1693 - 0.0052^{*}x + 5.2e\text{-}05^{*}x^2$; $R^2 = 0.90$. Black non-Hispanic: $y = 0.2948 - 0.0067^{*}x + 5.8e\text{-}05^{*}x^2$; $R^2 = 0.87$. Hispanic: $y = 0.2873 - 0.0083^{*}x + 8.5e\text{-}05^{*}x^2$; $R^2 = 0.95$. White non-Hispanic: $y = 0.1219 - 0.0021^{*}x + 1.5e\text{-}05^{*}x^2$; $R^2 = 0.92$. SI conversion factor: 1 ng/mL = 0.400641 nmol/L.

analysis indicated strong correlation between 25(OH)D level and SARS-CoV-2 positivity in all these groups (Fig 3A and 3B).

The association between lower SARS-CoV-2 positivity rates and higher circulating 25(OH)D levels per ng/mL remained significant in a multivariable logistic model (adjusted odds ratio 0.984, 95% C.I. 0.983–0.986; p<0.001). Other significant factors in both the adjusted and unadjusted models were male sex, northern and central latitudes, predominately black non-Hispanic zip codes, and predominately Hispanic zip codes (Table 1).

## Discussion

These results demonstrate an inverse relationship between circulating 25(OH)D levels and SARS-CoV-2 positivity. For the entire population those who had a circulating level of 25(OH)D <20 ng/mL had a 54% higher positivity rate compared to those who had a blood level of 30–34 ng/mL. The risk of SARS-CoV-2 positivity continued to decline until the serum levels reached 55 ng/mL. This finding is not surprising, given the established inverse relationship between risk of respiratory viral pathogens, including influenza, and 25(OH)D levels [14–16]. Vitamin D supplementation may reduce acute respiratory infections, especially in people with vitamin D deficiency [17]. A previous study found that each 4 ng/mL increase in circulating 25(OH)D levels was associated with a 7% decreased risk of seasonal infection, a decrement of approximately 1.75% per ng/mL [18]. This is remarkably similar to the 1.6% lower risk of SARS-CoV-2 positivity per ng/mL found in our adjusted multivariable model.

Patients with the lowest circulating levels of 25(OH)D had approximately 5–7% higher absolute SARS-CoV-2 positivity across northern, central, and southern latitudes. Indeed, Covid-19 diagnoses and particularly mortality exhibit a decreasing worldwide North-South latitude gradient [19]. The inverse relationship between SARS-CoV-2 positivity and 25(OH)D levels was most striking in predominately black non-Hispanic zip codes, followed by predominately Hispanic zip codes. Although 25(OH)D levels appeared to play a role for all race/ethnicities, patients from predominately black non-Hispanic zip codes had higher SASR-CoV-2 positivity than those from predominately white non-Hispanic zip codes at every 25(OH)D level. Other potential reasons including chronic diseases, occupations, housing, and lifetime risk exposures for the increased impact of COVID-19 on African Americans and Latinos have been published previously [20–22].

Northern and central latitudes, predominately Hispanic zip codes, predominately black non-Hispanic zip codes, age <60 years, and male sex were independently associated with both 25(OH)D levels and SARS-CoV-2 positivity. Yet, both in stratified analyses and in a model that controlled for all of these factors, the relationship between SARS-CoV-2 positivity and circulating 25(OH)D levels remained.

Limitations of this retrospective study include that testing for SARS-CoV-2 was based on selection factors, including presence and gravity of symptoms and exposure to infected individuals. High-risk groups, such as healthcare workers and first responders, are also more likely to be tested. Another limitation is that race/ethnicity estimates were based on aggregate U.S. Census proportions by zip code. There may be many other potentially confounding factors that were neither identified nor controlled for in this study. As expected, the multivariable model displayed poor overall fit and correlation statistics, given SARS-CoV-2 can infect

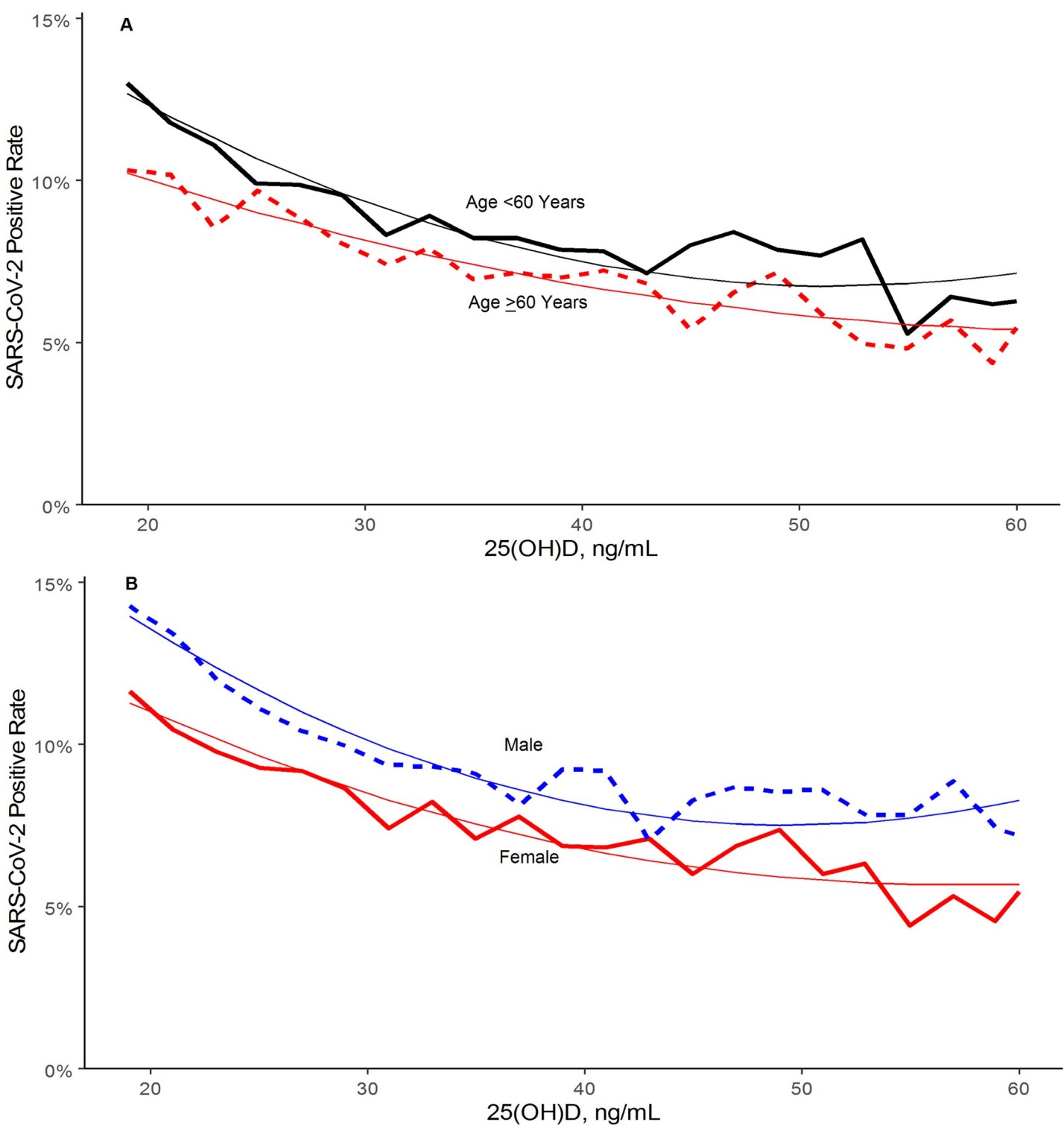

**Fig 3.** SARS-CoV-2 NAAT Positivity Rates and Circulating 25(OH)D Levels by (A) Age Group and (B) Sex. Smooth lines represent the weighted second order polynomial regression fit to the data associating circulating 25(OH)D levels (x) and SARS-CoV-2 positivity rates (y) where: Age <60: $y = 0.2161 - 0.0058^*x + 5.6e\text{-}05^*x^2$; $R^2 = 0.94$. Age $\geq$60: $y = 0.1515 - 0.0030^*x + 2.4e\text{-}05^*x^2$; $R^2 = 0.91$. Female: $y = 0.1837 - 0.0045^*x + 3.9e\text{-}05^*x^2$; $R^2 = 0.94$. Male: $y = 0.2445 - 0.0068^*x + 6.9e\text{-}05^*x^2$; $R^2 = 0.94$. SI conversion factor: 1 ng/mL = 0.400641 nmol/L.

**Table 1. Associations with SARS-CoV-2 positivity.**

|  | Unadjusted Odds Ratio (95% C.I.) | Adjusted Odds Ratio (95% C.I.) |
|---|---|---|
| 25(OH)D (per ng/mL increment) | 0.979 (0.977–0.980) | 0.984 (0.983–0.986) |
| Male | 1.26 (1.22–1.31) | 1.24 (1.20–1.28) |
| Female | reference | reference |
| Age ≥60 years | 0.74 (0.71–0.76) | 0.84 (0.81–0.87) |
| Age <60 years | reference | reference |
| *Latitudes* |  |  |
| Northern (>40 degrees) | 2.43 (2.32–2.54) | 2.66 (2.54–2.79) |
| Central (32–40 degrees) | 1.17 (1.12–1.23) | 1.22 (1.16–1.28) |
| Southern (<32 degrees) | reference | reference |
| *Race/Ethnicity zip codes* |  |  |
| Predominately black non-Hispanic | 2.04 (1.93–2.17) | 2.03 (1.91–2.15) |
| Predominately Hispanic | 1.61 (1.54–1.67) | 1.95 (1.87–2.04) |
| All other zip codes | reference | reference |

Adjusted model H-L Fit: p = 0.003; $R^2$ = 0.024. SI conversion factor: 1 ng/mL = 0.400641 nmol/L. Adjusted model included 188,028 patients with no missing values (98% of included patients).

anyone. The intent of the model was to determine whether circulating 25(OH)D levels remained significantly associated with SARS-CoV-2 positivity after adjustment for other identified factors.

The major strength of this study is the direct assessment of circulating 25(OH)D levels in a large cohort; this approach can more clearly elucidate the relationship of circulating 25(OH)D levels to SARS-CoV-2 positivity than is possible when using latitude as a surrogate for vitamin D status.

In conclusion, SARS-CoV-2 NAAT positivity is strongly and inversely associated with circulating 25(OH)D levels, a relationship that persists across latitudes, races/ethnicities, sexes, and age ranges. Our findings provide further rationale to explore the role of vitamin D supplementation in reducing the risk for SARS-CoV-2 infection and COVID-19 disease. If controlled trials find this relationship to be causative, the implications are vast and would present a cheap, readily-available method for helping prevent infection, especially for those with vitamin D deficiency. This could be of increased importance for the African American and Latinx community, who are disproportionately affected by both COVID-19 and vitamin D deficiency. In the interim, the authors recommend responsible vitamin D supplementation based on personal needs, risk factors, and advice from personal physicians in accordance with existing Endocrine Society Guidelines [23].

## Author Contributions

**Conceptualization:** Harvey W. Kaufman, Justin K. Niles, Martin H. Kroll, Caixia Bi, Michael F. Holick.

**Data curation:** Harvey W. Kaufman, Justin K. Niles, Martin H. Kroll, Caixia Bi.

**Formal analysis:** Harvey W. Kaufman, Justin K. Niles, Martin H. Kroll, Caixia Bi, Michael F. Holick.

**Investigation:** Justin K. Niles, Martin H. Kroll, Caixia Bi, Michael F. Holick.

**Methodology:** Harvey W. Kaufman, Justin K. Niles, Martin H. Kroll, Caixia Bi, Michael F. Holick.

**Project administration:** Harvey W. Kaufman, Michael F. Holick.

**Software:** Harvey W. Kaufman, Justin K. Niles, Martin H. Kroll, Caixia Bi.

**Supervision:** Michael F. Holick.

**Validation:** Michael F. Holick.

**Visualization:** Harvey W. Kaufman, Justin K. Niles, Martin H. Kroll, Caixia Bi, Michael F. Holick.

**Writing – original draft:** Harvey W. Kaufman, Justin K. Niles, Martin H. Kroll, Caixia Bi, Michael F. Holick.

**Writing – review & editing:** Harvey W. Kaufman, Justin K. Niles, Martin H. Kroll, Caixia Bi, Michael F. Holick.

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
