## [Decision Letter · Decision Letter 0]

19 Aug 2020

PONE-D-20-23650

SARS-CoV-2 Positivity Rates Associated with Circulating 25-Hydroxyvitamin D Levels

PLOS ONE

Dear Dr. Holick,

Thank you for submitting your manuscript to PLOS ONE. After careful consideration, we feel that it has merit but does not fully meet PLOS ONE’s publication criteria as it currently stands. Therefore, we invite you to submit a revised version of the manuscript that addresses the points raised during the review process.

ACADEMIC EDITOR:   The manuscript findings are interesting investigating the role of vitamin D supplementation in reducing the risk of SARS-CoV-2 infection. SARS-CoV-2 infection rate and vitamin D levels can be confounded by the assay method used.  Therefore, the authors should clarify the methods for sensitivity of assay and address the reviewers comments.

We look forward to receiving your revised manuscript.

Kind regards,

Dr. Sakamuri V. Reddy

Academic Editor

PLOS ONE

Journal Requirements:

2. In your ethics statement in the Methods section and in the online submission form, please provide additional information about the data used in your retrospective study. Specifically, please ensure that you have discussed whether all data were fully anonymized before you accessed them and/or whether the IRB or ethics committee waived the requirement for informed consent. If patients provided informed written consent to have data from their medical records used in research, please include this information.

'HWK, JKN, MHK, and CB are employees of Quest Diagnostics. HWK, MHK and CB

own stock in Quest Diagnostics. MFH is a consultant to Quest Diagnostics and was on the speakers’ bureau for Abbott Inc. and Hyatt Pharmaceutical Industries Company

PLC.'

We note that one or more of the authors are employed by a commercial company: Quest Diagnostics.

Additional Editor Comments (if provided):

Reviewers' comments:

Reviewer's Responses to Questions

**Comments to the Author**

1. Is the manuscript technically sound, and do the data support the conclusions?

Reviewer #1: Partly

Reviewer #2: Partly

2. Has the statistical analysis been performed appropriately and rigorously? 

Reviewer #1: No

Reviewer #2: No

3. Have the authors made all data underlying the findings in their manuscript fully available?

Reviewer #1: Yes

Reviewer #2: No

4. Is the manuscript presented in an intelligible fashion and written in standard English?

Reviewer #1: Yes

Reviewer #2: Yes

5. Review Comments to the Author

Reviewer #1: In this manuscript, Kaufman, et al. examined the relationship between the SARS-CoV-2 positivity and circulating levels of 25-hydroxyvitamin D (25OHD). They analyzed the data of from over 190,000 patients and found strong inverse correlation between SARS-CoV-2 positivity and 25OHD levels, which persisted across latitudes, races/ethnicities, both sexes and age ranges. This paper may provide a rationale to investigate the role of vitamin D supplementation in reducing the risk of SARS-CoV-2 infection. The reviewer’s specific comments are described below.

Specific comments

1. Figures and line 112-113 in the text. To analyze the relationship between circulating 25OHD levels and SARS-CoV-2 positivity, the authors assigned the 25OHD values <20 ng/mL or ≥�60 ng/mL as 19 ng/mL or 60 ng/mL, respectively. However, since many subjects have 25OHD levels lower than 20 ng/mL in the United States, the authors should use the raw data of 25OHD for the analyses.

2. Lines 87-91. Four different assays were used as SARS-Cov-2 RNA NAATs. Please provide the information on the sensitivity of each assay.

3. Table 1. “Vitamin D” should be changed to “25OHD”.

Reviewer #2: Kaufman et al. implemented retrospective analytic methods to identify an association between vitamin D levels and SARS-CoV-2 infection rate. The results are potentially interesting due to the need for viable SARS-CoV-2 treatments and a greater understanding for SARS-CoV-2 infection pathology. he manuscript is well-written and the figures are clear. However, interpreting these results are difficult due these vitamin D detection methods. Here, vitamin D was measured using two techniques: (1) immunoassay and (2) LC-MS. While LC-MS is considered the gold standard due to both its sensitivity and reliability, it is readily established that vitamin D immunoassays frequently overestimate or underestimate 25(OH)D concentrations (Kocak et al., 2015; Holmes et al., 2013; Farrell et al., 2012). I acknowledge the authors' efforts to consider all potential confounding variables related to ethnicity, geographical location, sex, and age. Nevertheless,the authors do not consider the vitamin D detection method as a limitation. Therefore, the association between SARS-CoV-2 infection rate and vitamin D levels may be confounded by the detection method. The regression analysis in which infection rate was assessed as a function of vitamin D levels appear positively skewed towards extremely low vitamin D levels, which could be due to the method of vitamin D detection.

In order to properly assess SARS-CoV-2 infection rate as a function of vitamin D levels, I suggest the authors to conduct analyses by detection method.

6. PLOS authors have the option to publish the peer review history of their article (what does this mean?). If published, this will include your full peer review and any attached files.

Reviewer #1: No

Reviewer #2: No

---

## [Author Response · Author response to Decision Letter 0]

28 Aug 2020

I confirm the following proposed Data Availability statement is accurate and suitable to appear alongside our manuscript.

"Data underlying the study cannot be made publicly available due to ethical concerns about patient confidentiality. Data will be made available to qualified researchers on request to HealthTrends@QuestDiagnostics.com."

Sincerely yours,

Michael F Holick PhD MD

Thank you for submitting your manuscript entitled "SARS-CoV-2 Positivity Rates Associated with Circulating 25-Hydroxyvitamin D Levels" to PLOS ONE. Your manuscript files have been checked in-house but before we can proceed we need you to address the following issues:

(1) Thank you for providing an updated Funding statement. We've made a minor change to meet our requirements, can you please confirm whether the following proposed statement is accurate and suitable to appear alongside your manuscript?

"Quest Diagnostics provided support in the form of salaries for authors JKN, BC, MHK, and HWK and consulting fees for MFH but did not have any additional role in the study design, data collection and analysis, decision to publish, or preparation of the manuscript. The specific roles of these authors are articulated in the ‘author contributions’ section.”

We will update the statement on your behalf with your approval.

Author’s response: Thank you, we approve, please provide middle initial (MHK) for author as well.

(2) We're also requesting the following update to your Competing Interest statement:

"HWK, JKN, MHK, and CB are employees of Quest Diagnostics. HWK, MHK and CB own stock in Quest Diagnostics. MFH is a consultant to Quest Diagnostics and was on the speakers’ bureau for Abbott Inc. and Hyatt Pharmaceutical Industries Company PLC. There are no patents, products in development or marketed products associated with this research to declare. This does not alter our adherence to PLOS ONE policies on sharing data and materials."

If any of this information is incorrect, please clarify, otherwise we will update the statement with your approval.

Author’s Response: Please see the response below for our statement on data sharing. If this conforms to your policies then the updated competing interest statement is approved. 

(3) We note that you've stated the following regarding your data: "The data underlying the results presented in the study are available from Quest Diagnostics Clinical Laboratory Database and are stored in the Quest Diagnostics Informatics Data Warehouse.”

PLOS only allows data to be available upon request if there are legal or ethical restrictions on sharing data publicly. For information on unacceptable data access restrictions, please see https://nam04.safelinks.protection.outlook.com/?url=http%3A%2F%2Fjournals.plos.org%2Fplosone%2Fs%2Fdata-availability%23loc-unacceptable-data-access-restrictions&data=02%7C01%7CHarvey.W.Kaufman%40questdiagnostics.com%7C204b422f20524f74c4bd08d849bc2198%7Cb68c6481b22b46b38c4c0024bb9b9b1f%7C1%7C0%7C637340419657309411&sdata=7dkabK3hbEyGDtegn0XYB5rParh4lkIFSbeny812%2Bsc%3D&reserved=0.

b) If there are no restrictions, please upload the minimal anonymized data set necessary to replicate your study findings to a stable, public repository and provide us with the relevant URLs, DOIs, or accession numbers. Please see https://nam04.safelinks.protection.outlook.com/?url=http%3A%2F%2Fwww.bmj.com%2Fcontent%2F340%2Fbmj.c181.long&data=02%7C01%7CHarvey.W.Kaufman%40questdiagnostics.com%7C204b422f20524f74c4bd08d849bc2198%7Cb68c6481b22b46b38c4c0024bb9b9b1f%7C1%7C0%7C637340419657309411&sdata=1tq7CqcqrkL%2F6Ke9QspE%2FQzIGfCmtiNp9rs1tYyFN9I%3D&reserved=0 for guidelines on how to de-identify and prepare clinical data for publication. For a list of recommended repositories, please see https://nam04.safelinks.protection.outlook.com/?url=https%3A%2F%2Fjournals.plos.org%2Fplosone%2Fs%2Frecommended-repositories&data=02%7C01%7CHarvey.W.Kaufman%40questdiagnostics.com%7C204b422f20524f74c4bd08d849bc2198%7Cb68c6481b22b46b38c4c0024bb9b9b1f%7C1%7C0%7C637340419657309411&sdata=Og4xLiA%2Fo0ERih%2Byo5fBPwpPWaRkTX3XmJRFeu%2BwJFU%3D&reserved=0. You also have the option of uploading the data as Supporting Information files, but we would recommend depositing data directly to a data repository if possible.

Authors’ Response: In accordance with Quest Diagnostics company policy on data security we are not able to upload the 191,000 rows of data to a public repository. We cannot violate HIPAA and open our Data Informatics Warehouse to external parties. This legal restriction applies to academic institutions as well as to our commercial organization.

Quest Diagnostics has previously published in PLoSONE: 

https://journals.plos.org/plosone/article?id=10.1371/journal.pone.0180840

https://journals.plos.org/plosone/article?id=10.1371/journal.pone.0118108

https://journals.plos.org/plosone/article?id=10.1371/journal.pone.0063416

https://journals.plos.org/plosone/article?id=10.1371/journal.pone.0028201

In all prior circumstances as with this study, we have made data available to researchers upon request. We are eager to have others validate our analysis, suggest new ways to analyze and interpret our data, and explore how we can further understanding of health and disease. In recent months we have worked on other subjects with investigators from many outside organizations including the CDC, Boston University, Penn State University, the University of Alabama, the Alameda Health System, and many others. Requests for data and research collaborations can be sent to HealthTrends@QuestDiagnostics.com

(4) In the Methods section of your manuscript, please include the following statement: "HIPAA clearly defines research use of data as analyzed for this and numerous other studies based on the Quest Diagnostics Data Informatics Warehouse (45 CFR 164.501, 164.508, 164.512(i) (See also 45 CFR 164.514(e), 164.528, 164.532) Link: https://nam04.safelinks.protection.outlook.com/?url=https%3A%2F%2Fwww.hhs.gov%2Fhipaa%2Ffor-professionals%2Fspecial-topics%2Fresearch%2Findex.html&data=02%7C01%7CHarvey.W.Kaufman%40questdiagnostics.com%7C204b422f20524f74c4bd08d849bc2198%7Cb68c6481b22b46b38c4c0024bb9b9b1f%7C1%7C0%7C637340419657314405&sdata=R9Yjj38kKasss73QqHM0u2z16rLwk64VX3QAImAFrRg%3D&reserved=0 ). Quest Diagnostics takes the additional step of having its process reviewed annual by the Western Institutional Review Board who has determined the process is “deemed exempt."

Authors’ Response: We have added this statement to the methods section. 

Your manuscript has been returned to your account. Please log on to PLOS Editorial Manager at https://nam04.safelinks.protection.outlook.com/?url=https%3A%2F%2Fwww.editorialmanager.com%2Fpone%2F&data=02%7C01%7CHarvey.W.Kaufman%40questdiagnostics.com%7C204b422f20524f74c4bd08d849bc2198%7Cb68c6481b22b46b38c4c0024bb9b9b1f%7C1%7C0%7C637340419657314405&sdata=%2F2Lb3ZZU82hslYEkPeYh8KeqjCgaz5Yz8r9Olq47s%2BU%3D&reserved=0 to access your manuscript.

Your manuscript can be found in the "Revisions Sent Back to the Author" link under the New Submissions menu. After you have made the changes requested above, please be sure to view and approve the revised PDF after rebuilding the PDF to complete the resubmission process.

Please note that these changes have been requested to comply with submission guidelines and your manuscript will *not* be sent to review until you have fully adhered to our requests. Once your paper has been seen by an Editor we may return it to you for further information or amendments.

We ask that you address this request within 28 days. If you require additional time, please email the journal office. We are happy to grant extensions of up to one month past this due date. If we have not heard from you within 28 days, your manuscript will be withdrawn from Editorial Manager.

Kind regards,

Agnes Magyar

PLOS ONE

Journal Requirements:

Authors’ Reponse: We have made the formatting changes outlined in these documents.

2. In your ethics statement in the Methods section and in the online submission form, please provide additional information about the data used in your retrospective study. Specifically, please ensure that you have discussed whether all data were fully anonymized before you accessed them and/or whether the IRB or ethics committee waived the requirement for informed consent. If patients provided informed written consent to have data from their medical records used in research, please include this information.

Authors’ response: HIPAA clearly defines research use of data as analyzed for this and numerous other studies based on the Quest Diagnostics Data Informatics Warehouse (45 CFR 164.501, 164.508, 164.512(i) (See also 45 CFR 164.514(e), 164.528, 164.532) Link: https://www.hhs.gov/hipaa/for-professionals/special-topics/research/index.html ). Quest Diagnostics takes the additional step of having its process reviewed annual by the Western Institutional Review Board who has determined the process is “deemed exempt.” This statement is included in the manuscript text to assure the readers. Finally, patient privacy and adherence to HIPAA is of prime important to Quest Diagnostics to provide patients, healthcare providers, and the public trust in its operations ( https://www.questdiagnostics.com/home/privacy/ ). 

'HWK, JKN, MHK, and CB are employees of Quest Diagnostics. HWK, MHK and CB

own stock in Quest Diagnostics. MFH is a consultant to Quest Diagnostics and was on the speakers’ bureau for Abbott Inc. and Hyatt Pharmaceutical Industries Company

PLC.'

We note that one or more of the authors are employed by a commercial company: Quest Diagnostics.

Authors’ Response: We added: “The funder provided support in the form of salaries for authors JKN, BC, MK, and HWK and consulting fees for MFH but did not have any additional role in the study design, data collection and analysis, decision to publish, or preparation of the manuscript. The specific roles of these authors are articulated in the ‘author contributions’ section.”

Authors’ response: Not applicable.

Authors’ Response: The Competing Interests Statement is in compliance with this request. We will add the following to the data sharing statement: “The data underlying the results presented in the study are available from Quest Diagnostics Clinical Laboratory Database and are stored in the Quest Diagnostics Informatics Data Warehouse.” 

Authors’ Response: We have added these to an updated cover letter.

Additional Editor Comments (if provided):

Reviewers' comments:

Reviewer's Responses to Questions

Comments to the Author

1. Is the manuscript technically sound, and do the data support the conclusions?

Reviewer #1: Partly

Reviewer #2: Partly

2. Has the statistical analysis been performed appropriately and rigorously? 

Reviewer #1: No

Reviewer #2: No

3. Have the authors made all data underlying the findings in their manuscript fully available?

Reviewer #1: Yes

Reviewer #2: No

4. Is the manuscript presented in an intelligible fashion and written in standard English?

Reviewer #1: Yes

Reviewer #2: Yes

5. Review Comments to the Author

Reviewer #1: In this manuscript, Kaufman, et al. examined the relationship between the SARS-CoV-2 positivity and circulating levels of 25-hydroxyvitamin D (25OHD). They analyzed the data of from over 190,000 patients and found strong inverse correlation between SARS-CoV-2 positivity and 25OHD levels, which persisted across latitudes, races/ethnicities, both sexes and age ranges. This paper may provide a rationale to investigate the role of vitamin D supplementation in reducing the risk of SARS-CoV-2 infection. The reviewer’s specific comments are described below.

Specific comments

1. Figures and line 112-113 in the text. To analyze the relationship between circulating 25OHD levels and SARS-CoV-2 positivity, the authors assigned the 25OHD values <20 ng/mL or ≥�60 ng/mL as 19 ng/mL or 60 ng/mL, respectively. However, since many subjects have 25OHD levels lower than 20 ng/mL in the United States, the authors should use the raw data of 25OHD for the analyses.

Authors’ Response: We chose the cutoff of <20 ng/mL to be more clinically relevant as it is in keeping with the definition of vitamin D deficiency. Given the small number of patients with extreme values, and our message that compares those with vitamin D deficiency to patients where the end of increasing benefits exists at 55 ng/mL we feel these cutoffs are appropriate. However, we acknowledge that most of the patients with vitamin D deficiency have values between 16-20 ng/mL, enough to be statistically relevant. If we were to analyze the data with the bottom bin at 15 ng/mL the following would happen:

The R-squared value in Figure 1 would increase from 0.96 to 0.97. The unadjusted odds ratio for 25(OH) would increase 0.001 from 0.979 (95% CI 0.977-0.980) to 0.980 (95% CI 0.978-0.981). The adjusted odds ratio for 25(OH) would also increase 0.001 from 0.984 (95% CI 0.983-0.986) to 0.985 (95% CI 0.984-0.986). All effect sizes of other factors in the multivariable model remain identical except the predominately black non-Hispanic ZIP codes, which would decrease 0.01. The mean 25(OH) level would fall from 31.7 (SD 11.7) to 31.2 (SD 12.3). 

Given the lack of clinically significant impact on the main statistical outcomes, the clinical relevance gained by grouping all vitamin D deficient patients, and the urgent nature of the message this study contains (a major reason we chose to submit to PLOS ONE, expediency) we would greatly prefer to keep the groupings unchanged. 

2. Lines 87-91. Four different assays were used as SARS-Cov-2 RNA NAATs. Please provide the information on the sensitivity of each assay.

Authors’ Response: We have added four FDA references for the four tests in question and added the following statement to the methods section: “We combined results from all four tests due to their very similar sensitivity and specificity.[8-11]”

3. Table 1. “Vitamin D” should be changed to “25OHD”.

Authors’ Response: We agree and have made this change. Thank you for catching this. 

Reviewer #2: Kaufman et al. implemented retrospective analytic methods to identify an association between vitamin D levels and SARS-CoV-2 infection rate. The results are potentially interesting due to the need for viable SARS-CoV-2 treatments and a greater understanding for SARS-CoV-2 infection pathology. The manuscript is well-written and the figures are clear. However, interpreting these results are difficult due these vitamin D detection methods. Here, vitamin D was measured using two techniques: (1) immunoassay and (2) LC-MS. While LC-MS is considered the gold standard due to both its sensitivity and reliability, it is readily established that vitamin D immunoassays frequently overestimate or underestimate 25(OH)D concentrations (Kocak et al., 2015; Holmes et al., 2013; Farrell et al., 2012). I acknowledge the authors' efforts to consider all potential confounding variables related to ethnicity, geographical location, sex, and age. Nevertheless,the authors do not consider the vitamin D detection method as a limitation. Therefore, the association between SARS-CoV-2 infection rate and vitamin D levels may be confounded by the detection method. The regression analysis in which infection rate was assessed as a function of vitamin D levels appear positively skewed towards extremely low vitamin D levels, which could be due to the method of vitamin D detection.

In order to properly assess SARS-CoV-2 infection rate as a function of vitamin D levels, I suggest the authors to conduct analyses by detection method.

Authors’ Response: 98.8% of specimens were tested using immunoassay. In a company-wide analysis the immunoassay and LC/MS-MS methods obtain nearly identical values, with the immunoassay yielding values 1 to 2 ng/mL less, on average, than the LC/MS-MS methodology. In addition, the difference between the methods is tiny in comparison to the range of results and if it did have an effect, it would only diminish the statistical significance of results, not magnify the effect. In the regression analysis in which infection rate was assessed as a function of 25(OH)D levels the specimens tested by LC/MS-MS had an unadjusted odds ratio of 0.956 (95% CI 0.942-0.970), stronger than the unadjusted odds ratio for immunoassay tests 0.979 (95% CI 0.977-0.980) which is also the results for all specimens combined.

We have added the following sentence to the results: “98.8% of included patients had 25(OH)D levels assessed with immunoassay testing methodology.”

6. PLOS authors have the option to publish the peer review history of their article (what does this mean?). If published, this will include your full peer review and any attached files.

Do you want your identity to be public for this peer review? For information about this choice, including consent withdrawal, please see our Privacy Policy.

Reviewer #1: No

Reviewer #2: No

---

## [Decision Letter · Decision Letter 1]

3 Sep 2020

SARS-CoV-2 Positivity Rates Associated with Circulating 25-Hydroxyvitamin D Levels

PONE-D-20-23650R1

Dear Dr. Holick,

We’re pleased to inform you that your manuscript has been judged scientifically suitable for publication and will be formally accepted for publication once it meets all outstanding technical requirements.

Kind regards,

Dr. Sakamuri V. Reddy

Academic Editor

PLOS ONE

Additional Editor Comments (optional):

Reviewers' comments:

Reviewer's Responses to Questions

**Comments to the Author**

1. If the authors have adequately addressed your comments raised in a previous round of review and you feel that this manuscript is now acceptable for publication, you may indicate that here to bypass the “Comments to the Author” section, enter your conflict of interest statement in the “Confidential to Editor” section, and submit your "Accept" recommendation.

Reviewer #1: All comments have been addressed

2. Is the manuscript technically sound, and do the data support the conclusions?

Reviewer #1: Yes

3. Has the statistical analysis been performed appropriately and rigorously? 

Reviewer #1: Yes

4. Have the authors made all data underlying the findings in their manuscript fully available?

Reviewer #1: Yes

5. Is the manuscript presented in an intelligible fashion and written in standard English?

Reviewer #1: Yes

6. Review Comments to the Author

Reviewer #1: The responses to the questions raised by this reviewer are satisfactory, and the manuscript has been appropriately revised.

7. PLOS authors have the option to publish the peer review history of their article (what does this mean?). If published, this will include your full peer review and any attached files.

Reviewer #1: No

---

## [Editor Report · Acceptance letter]

8 Sep 2020

PONE-D-20-23650R1 

SARS-CoV-2 positivity rates associated with circulating 25-hydroxyvitamin D levels 

Dear Dr. Holick:

I'm pleased to inform you that your manuscript has been deemed suitable for publication in PLOS ONE. Congratulations! Your manuscript is now with our production department. 

Kind regards, 

on behalf of

Dr. Sakamuri V. Reddy 

Academic Editor

PLOS ONE